# Detection of *TERT* Promoter Mutations as a Prognostic Biomarker in Gliomas: Methodology, Prospects, and Advances

**DOI:** 10.3390/biomedicines10030728

**Published:** 2022-03-21

**Authors:** Tsimur Hasanau, Eduard Pisarev, Olga Kisil, Naosuke Nonoguchi, Florence Le Calvez-Kelm, Maria Zvereva

**Affiliations:** 1Department of Chemistry, Lomonosov Moscow State University, 119991 Moscow, Russia; mrtimurgasanov@gmail.com; 2Faculty of Bioengineering and Bioinformatics, Lomonosov Moscow State University, 119234 Moscow, Russia; e.pisarev@fbb.msu.ru; 3Chair of Chemistry of Natural Compounds, Department of Chemistry, Lomonosov Moscow State University, 119991 Moscow, Russia; 4Gause Institute of New Antibiotics, 119021 Moscow, Russia; olvv@mail.ru; 5Department of Neurosurgery, Osaka Medical and Pharmaceutical University, Takatsuki 569-8686, Japan; naosuke.nonoguchi@ompu.ac.jp; 6Genomic Epidemiology Branch, International Agency for Research on Cancer (IARC), 69372 Lyon, France; lecalvezf@iarc.fr

**Keywords:** telomerase reverse transcriptase, telomerase activation, *TERT*, *TERT* promoter region, *TERT* mutations, glioma, central nervous system tumors, molecular biomarkers, noninvasive detection, dPCR, ddPCR, Sanger sequencing, NGS, MRI

## Abstract

This article reviews the existing approaches to determining the *TERT* promoter mutational status in patients with various tumoral diseases of the central nervous system. The operational characteristics of the most common methods and their transferability in medical practice for the selection or monitoring of personalized treatments based on the *TERT* status and other related molecular biomarkers in patients with the most common tumors, such as glioblastoma, oligodendroglioma, and astrocytoma, are compared. The inclusion of new molecular markers in the course of CNS clinical management requires their rapid and reliable assessment. Availability of molecular evaluation of gliomas facilitates timely decisions regarding patient follow-up with the selection of the most appropriate treatment protocols. Significant progress in the inclusion of molecular biomarkers for their subsequent clinical application has been made since 2016 when the WHO CNS classification first used molecular markers to classify gliomas. In this review, we consider the methodological approaches used to determine mutations in the promoter region of the TERT gene in tumors of the central nervous system. In addition to classical molecular genetical methods, other methods for determining *TERT* mutations based on mass spectrometry, magnetic resonance imaging, next-generation sequencing, and nanopore sequencing are reviewed with an assessment of advantages and disadvantages. Beyond that, noninvasive diagnostic methods based on the determination of the mutational status of the *TERT* promoter are discussed.

## 1. Introduction

Telomere length maintenance systems perform an essential function in preserving genome stability. Abnormalities in the functioning of these systems, such as telomerase reactivation, usually play a key role in the course of oncogenesis [1]. One of the mechanisms of telomerase reactivation in oncogenesis involves activation of the transcription of the main catalytic component of telomerase due to the occurrence of somatic mutations in the promoter region of the telomerase reverse transcriptase (TERT) gene [2]. These genomic changes have been reported in a wide range of tumor genomes [3], including in tumors of the central nervous system (CNS), where they have been observed at the highest frequency. The regulation of telomerase expression in gliomas has been shown to depend on the mutational status of the *TERT* promoter (*TERTp*) [4]. In 2016, the World Health Organization (WHO) CNS classification first applied molecular markers to classify gliomas, and the recent revised 2021 version reinforced the utility of these markers for more accurate CNS grading systems. Numerous molecular changes of clinicopathological significance are included in the WHO CNS5. In the WHO CNS5 classification system, the mutational status of the telomerase reverse transcriptase promoter is one of the key genetic markers of gliomas [5,6,7].

Cancer-specific TERT expression and mediated telomerase activation have always generated great enthusiasm for the potential clinical applications of TERT/telomerase-based assays in cancer. However, the unstable nature of TERT mRNA and telomerase RNA makes it challenging to reliably analyze direct *TERT* expression or telomerase activity for cancer diagnosis or monitoring. Nevertheless, numerous clinical research has positively evaluated telomerase activation, especially TERT-related changes, as prognostic factors for cancer patients [7]. However, the recent discovery of widespread *TERTp* mutations in various tumors provides new opportunities for simple and lost-cost biomarkers in detecting patients with *TERT*-mutated tumors. 

The integration of new molecular markers into routine diagnostics requires their rapid and reliable assessment. Currently, there are many genomic technologies that allow detecting the presence of mutations in tissues or bodily fluids. In this review, various methods for detecting mutations in the promoter region of the TERT gene are presented, their analytical performance compared, and their possible advantages and disadvantages for potential clinical implementation discussed. 

## 2. Results and Discussion

### 2.1. Classification of CNS Tumors

There are three main types of glial cells: astrocytes, oligodendrogliocytes, and ependymocytes. According to the type of cells from which a glioma of the brain originates, neurology distinguishes astrocytoma, oligodendroglioma, and ependymoma; there are also mixed gliomas of the brain, such as oligoastrocytomas [8]. According to the WHO classification, there are four grades (classes) of malignancy of gliomas in the brain [5]. Grade I gliomas, which are often considered benign, are usually curable by complete surgical resection and rarely, if ever, progress to higher-grade lesions. In contrast, grade II or III gliomas are invasive and progress to higher degrees of lesions. Diffuse grade II and III gliomas are usually less aggressive than higher grade tumors, with a median survival of more than seven years [8]. There is significant heterogeneity among grade II and III gliomas in terms of pathological features and clinical results. WHO grade IV tumors (glioblastomas), the most invasive form, have the worse prognosis, with a 5-year survival rate of less than 5% after initial diagnosis. After resection of primary glioblastoma, local recurrences may occur in the area of the removed tumor lesion due to the character of tumor growth and formation because of the presence of tumor cells in the adjacent pre-tumor area. Primary glioblastoma develops rapidly without preceding low-grade lesions, while secondary glioblastoma slowly progresses from diffuse or anaplastic astrocytoma (grades II and III according to the WHO classification). Primary and secondary glioblastoma differ genetically rather than histologically [5,9].

Different subtypes of gliomas have different aggressiveness spectra and responses to therapeutic treatments. The identification of a particular malignancy into a particular class has long been determined by histological characteristics supported by ancillary tissue-based tests (e.g., immunohistochemical, ultrastructural). However, diagnosis based on histology is subject to much variability between observers. In addition to the diagnostic problems, traditional diagnostic/classification schemes have fallen short of prognostic accuracy, even for patients with the same diagnosis (e.g., grade IV glioblastoma), where survival rates can vary from weeks or months to several years [9,10,11]. In addition, research performed in the last decade has shown that the impact of molecular genetic changes on disease outcome is more significant than some changes in the therapeutic treatment. The development of advanced molecular diagnostic techniques has led to a better understanding of the genomic drivers involved in gliomagenesis and their important prognostic values. The fifth edition of the 2021 WHO Classification of Primary CNS Tumors makes important changes that enhance the role of molecular diagnosis in the classification of CNS tumors [5,6]. For each tumor type, different molecular targeted markers are characterized. Taken/evaluated together these molecular markers allow the risk stratification of patients with CNS tumors in terms of prognosis and response to treatment.

### 2.2. Telomeres, Telomerase, and the TERT Promoter

Mutations of the *TERTp* are known to be absent in normal human cells but are often associated with malignant tumor progression and enhanced proliferation of cells. The two most common mutations in *TERTp* that are mutually exclusive in CNS tumors are C228T and C250T, located −124 and −146 bp, respectively, prior to the TERT transcription site (chr 51,295,228 C > T and 1,295,250 C > T, respectively, according to GRCh37.p13 genome assembly, 1,295,113 and 1,295,135 according to GRCh38.p13 assembly). The localization of these mutations on the *TERTp* is shown in Figure 1.

These *TERTp* mutations create new binding sites for E-26 family transcription factors tryptophan cluster factors class (ETS/TCF) and cause a two- to four-fold increase in transcription of the messenger RNA of the telomerase catalytic subunit [12]. In addition, among members of this family, mutated *TERTp* creates binding sites in CNS tumors for ETS1/2 [13,14] and GA-binding proteins (GABP) [12]. The binding sites for the ETS transcription factor were specified based on the JASPAR CORE database containing a set of profiles derived from published collections of experimentally determined transcription factor binding sites [15].

The main *TERT* promoter does not contain a TATA-box and a CAAT-box, but it includes an array of five GC-boxes surrounded by two E-boxes [16]. As a result, *TERTp* can form a G-quadruplex structure. G-quadruplexes of DNA (G4) are known to be a component of a complex regulatory system in both normal and pathological cells [17] and can complicate the detection of changes in the primary structure of DNA.

Somatic *TERTp* mutations have been observed in various forms of CNS tumors [18,19]. *TERTp* mutations in CNS tumors correlate with the presence of other biomarkers, such as tumor suppressor protein 53 (TP53) gene mutations, isocitrate dehydrogenase 1/2 (IDH1/IDH2) gene mutations, epidermal growth factor receptor (EGFR) changes resulting in overexpression, co-deletion of chromosomal arms 1p and 19q (1p/19q co-deletion), O6-methylguanine-DNA methyltransferase (MGMT) gene promoter methylation status (*MGMTp*), and nuclear alpha-thalassemia/mental retardation X-linked syndrome (ATRX) gene mutations [4]. According to Powter et al., gliomas of low malignancy had the highest rate of co-detection of *TERTp* and *IDH1/2* mutations (87.3%), while glioblastomas had a low frequency (joint detection of *TERTp* and *IDH1/2* mutations 11.5%). Tumors (*TERTp*-mut + *IDH*-wt) were significantly associated with *EGFR* amplification (44.1%). A total of 54.6% of low-grade gliomas, 71.4% of glioblastomas, and all anaplastic gliomas had *TERTp* and phosphatase and tensin homolog (*PTEN*) co-mutations. *TERTp*-mut and *MGMTp* hypermethylation accounted for 51% in low-grade gliomas and 43.6% in glioblastomas. *TERTp*-mut is identified in 87.9% of gliomas with a 1p/19q co-deletion. *TERTp*-mut is associated with the suppressor/enhancer of Lin-12-like (SEL1L) overexpression. All parameters correlate with overall survival (OS) prognosis [4].

Mutations in *TERTp* causing an increase in telomerase activity and telomere elongation are observed in both the most aggressive form of diffuse glioma, astrocytoma, and the least aggressive form, oligodendroglioma. Hence, telomere maintenance may be a necessary precondition for the formation of CNS tumors [20]. Considering the use of *TERTp* mutations as a diagnostic or prognostic marker of CNS tumors is therefore highly relevant. 

### 2.3. Telomere Length as a Prognostic Factor for Patients with CNS Tumors

Indirectly, mutations in the TERT gene promoter lead to telomere elongation. Gao and colleagues [21] measured the relative telomere length of 23 grade I gliomas, which are considered “borderline tumors” (most scientists believe it can be cured after surgery) and showed that telomere length had a significant impact on the survival prognosis for patients. None of the eight patients with short telomere tumor cells died, versus 6 of 15 (40%) patients with long telomeres (died). These results confirm that telomere length may be an important predictor of clinical results in low-grade gliomas of malignancy patients. According to these results, longer telomeres are more typical for gliomas than for meningiomas. *TERTp* mutations and longer telomeres were predictors of worse survival for glioma patients regardless of gender, age, severity, *IDH1* and *MGMTp* status, radiation therapy, and chemotherapy. Co-detected *TERTp* mutations (*TERTp*-mut) and telomere elongation are associated with a worse prognosis in patients more frequently than those detected individually. Notably, patients with *TERTp-*mut, especially those with C228T, or patients with elongated telomeres, were resistant to radiotherapy. Gao and colleagues revealed that telomere length was significantly shorter in *TERTp*-mut cases than in cases with an unchanged promoter sequence (*TERTp-*wt) [21]. Heidenreich and colleagues also showed that telomere length is shorter in gliomas with *TERTp*-mut compared to gliomas without *TERTp*-mut [22].

Whether *TERTp* mutations are an early or late event in the genesis of CNS tumors has not yet been fully clarified and requires additional investigations. A recent analysis of patients with bladder cancer showed that mutations in *TERTp* could be detected in urine samples ten years before the initial clinical diagnosis of bladder cancer. These mutations were absent among comparable control groups that did not develop cancer for 10 years after sampling [23]. Wang and colleagues detected the mutation in both benign follicular thyroid adenoma and precancerous lesions or follicular tumors with atypical thyroid adenoma or uncertain malignancy potential [24,25]. The frequency of *TERTp* mutations in the above precancerous lesions was 17%, the same as that of its fully transformed analog of follicular thyroid carcinoma. Importantly, all mutation-carrying malignancies discussed express TERT mRNA and exhibit telomerase activity, which proves that telomerase reactivation occurs within early oncogenic thyroid lesions. Similarly, *TERTp* mutation has been identified as an early event in transforming precancerous hepatocellular carcinoma nodules in liver cirrhosis, melanoma, and urothelial papilloma [26,27,28]. At present, it is unclear exactly how *TERTp* mutations occur in early oncogenic lesions and completely transformed cells. Although the TERT overexpression level is typically observed in tumors, those with *TERTp*-mut have shorter telomeres than non-cancerous tissues. This fact indicates that *TERTp* mutations can represent a later event in oncogenesis (second phase) when telomere length has already been depleted [17,18]. Another hypothesis suggests that somatic mutations in cells are accumulating at a constant rate throughout life [29]. Whole genome sequencing data analysis of the Cancer Genome Atlas data of adult diffuse glioma did not show that *TERTp* mutations are associated with increased telomere length in grade II–III–IV diffuse gliomas [30] which argues for an early oncogenic step in this lesion. Abou and colleagues suggested that glioblastoma develops early from a common precursor with loss of at least one copy of the PTEN gene (heterozygous deletion) and a *TERTp* mutation: this assumption is based on the high frequency of their co-detection in gliomas [31].

### 2.4. Mutation Status of the TERT Promoter as a Prognostic Marker

The latest edition of the WHO Classification of Primary CNS Tumors in 2021 defined changes in the promoter region of the TERT gene as one of the key molecular diagnostic markers in the classification of CNS tumors for their treatment [5,6,7]. In three types of primary tumors (oligodendroglioma, glioblastoma, and meningioma), *TERTp*-mut is one of the diagnostic parameters. Furthermore, the combined detection of *TERTp*-mut and *IDH1/2*-mut is considered an alternative feature of oligodendroglioma.

*TERTp* mutations are the most frequent genomic changes in CNS tumors. Arita and colleagues investigated the presence of mutations in *TERTp* in a series of 546 gliomas [19]. They found a high frequency of mutually exclusive mutations located at common sites, C228T and C250T in all subtypes of the analyzed CNS tumors, of different classes in an average of 55% of all cases. The frequency of mutations was particularly high among primary glioblastomas (70%) and oligodendrogliomas (74%) but relatively low among diffuse astrocytomas (19%) and anaplastic astrocytomas (25%). A similar percentage distribution was shown by a meta-analytic approach (bibliography search) carried out in 2016: *TERTp*-mut was frequently found in glioblastoma (69%) and oligodendroglioma (72%), but less frequently in astrocytomas (24%) and oligoastrocytomas (38%) [32]. Other research has also evaluated the incidence of *TERT* mutations in different types of gliomas. Based on these data, *TERT* mutations are the most frequently found in glioblastoma (WHO grade IV), oligodendroglioma (WHO grade II), and oligoastrocytoma (WHO grade II), and are also frequently found in diffuse astrocytoma (WHO grade II), anaplastic astrocytoma (WHO grade III), anaplastic oligoastrocytoma (WHO grade III), and anaplastic oligodendroglioma (WHO grade III) [19,22,32,33,34,35,36,37,38]. The data are summarized in Table 1. In comparison to CNS tumors in adults, *TERTp* mutations were exceedingly rare in tumors typically encountered in pediatric patients [39].

According to the data presented in Table 1, the molecular profiles of low- and high-grade gliomas have different frequencies of *TERTp* mutations regardless of tumor class. For example, the highest frequency of *TERTp* mutations was found in glioblastomas (WHO grade IV) with an average frequency of 70%, oligodendrogliomas (WHO grade II) with an average frequency of 77%, and anaplastic oligodendrogliomas (WHO grade III) with an average frequency of 74%. At the same time, the lowest frequency of *TERTp* mutations was found in diffuse astrocytoma (WHO grade II) with an average frequency of 25%, anaplastic astrocytoma (WHO grade III) with an average frequency of 31%, oligoastrocytoma (WHO grade II) with an average frequency of 46%, and anaplastic oligoastrocytoma (WHO grade III) with an average frequency of 42% [19,22,32,33,34,35,36,37,38].

The most frequent point mutation among gliomas with *TERTp*-mut was C228T, in ¾ of the cases. Although C228T and C250T mutations have been reported to be mutually exclusive in CNS tumors, Nonoguchi and colleagues found C228T and C250T co-mutations in only 1 case among 322 *IDH*-wt glioblastomas in which mutations in both sites were found, with a frequency of 0.31% (1/322). In this study, authors examined *TERTp*-mut in C228T and C250T using a cohort of 358 glioblastoma cases in a population-based study that included 36 *IDH*-mut glioblastoma cases [40]. The fact that C228T and C250T mutations are mutually exclusive in gliomas suggests that they are each individually sufficient to play a significant oncogenic role in the pathogenesis of gliomas. 

Both C228T and C250T mutations generate identical sequences, provide ETS family transcription factor binding, and are equally effective in enhancing TERT transcription. In vivo, the −124 C > T mutation was associated with higher TERT expression in glioblastoma [12]. This may indicate that the ETS/TCF binding site at the −124 position provides a more favorable/available access point for the transcriptional machinery [12]. Thus, despite similar far-reaching effects, the two canonical *TERTp* mutations can distinctly alter the biology of TERT expression. The mechanism(s) mediating the induction of TERT transcription in cells carrying these mutations remains poorly understood. Perhaps the two *TERTp* mutations generating the same ETS binding site are functionally different in the sense that C250T, unlike C228T, is similarly controlled by noncanonical NF-κB signal transduction [13].

It is also known that these mutations are absent in benign tumors and in tissues of healthy individuals [2]. Akyerli et al. identified several clinical correlations of *TERTp*-mut in patients with gliomas [41]. Mutations were present in more than half (52.7%) of patients, and *TERTp*-mut C228T patients showed lower OS compared to *TERTp*-mut C250T patients. *TERTp*-muts were found to be homogeneously present in the tumors but not in the surrounding brain parenchyma. *TERTp*-mut tumor status did not change over time despite adjuvant therapy or recurrence. The above allows considering *TERTp* mutation status as a reliable diagnostic and prognostic factor for CNS tumors. 

Hewer and colleagues proposed a technique combining *IDH1/IDH2* and *TERT**p* (C228T, C250T) mutations assays to distinguish diffuse gliomas from reactive gliosis. The *TERTp* mutation assay was successfully applied to distinguish gliomas from gliosis for older adults. *TERTp* mutations were not detected in any of the 58 (0%) reactive gliosis samples and in 91 of 117 (78%) *IDH* wild-type gliomas. Furthermore, based on a series of 200 consecutive diffuse gliomas, they found that the *IDH* mutation assay only had a sensitivity of 28% to detect gliomas, whereas the combined assay yielded a sensitivity of 85% [42].

A correlation between the occurrence of *TERTp* mutation and OS in patients with glioma was reported [34,41]. Generally, for patients with high malignancy glioma, the group with *TERTp*-mut has a significantly worse OS compared to the *TERTp*-wt group. However, gliomas harboring *TERTp* mutations are often classified as grade IV gliomas because they initially have a worse predicted OS: only 39 tumors out of 406 (9.6%) in the Eckel-Passow and colleagues research were grade II or III [20]. When the cohort was considered only for glioblastoma (the most aggressive form of glioma), the following was observed: patients with *TERTp*-mut had a shorter OS (11 months) compared to patients with *TERTp*-wt (20 months) [43]. Nonoguchi and colleagues showed that *TERTp*-mut status had no effect on OS in glioblastomas when adjusted for other genetic changes and that the prognostic value of *TERTp* mutations was largely due to their inverse correlation with *IDH1* mutations [40]. In low-grade gliomas, the prognostic value of the *TERTp* mutation clearly depends on the mutational status of *IDH1/2*. Yang and colleagues reported that the *TERTp* mutation is a prognostic factor for good OS in grade II/III gliomas, 70–90% of which harbor *IDH* mutations [35]. Some inconsistency in the assessment of the prognostic value of *TERTp* mutations may be due to insufficient cohort size or different treatment procedures in the evaluated cohorts. For example, the presence of *TERTp*-mut is strongly associated with diagnosis at an older age, which in itself is a well-known prognostic factor and influences treatment decisions.

Summarizing this section, can *TERTp* mutation status be considered an independent biomarker of primary glioma? Currently, this question cannot be answered definitively. A potential negative independent prognostic impact of *TERTp* mutations was identified; the deleterious effect of *TERTp*-mut is correlated with the presence of associated molecular and clinical factors, such as older age, *IDH*-wt status, and *MGMTp* hypermethylation [44]. On the other hand, it is noteworthy that *TERTp* mutations are a significant prognostic marker in other cancers (e.g., melanoma, thyroid cancer, urothelial carcinoma) and are independent of other mutations. The currently known data show that the prognostic impact of the presence of *TERTp*-mut in CNS tumors depends largely on the context of the histological and genomic background of the tumor, primarily the *IDH* status [45], and the methodology for determining mutations in the promoter region of the TERT gene.

### 2.5. Analysis of Methods to Detect TERT Promoter Mutations in CNS Tumors

#### 2.5.1. Sanger Sequencing

At first, detection of the C228T and C250T *TERTp* mutations was performed using routine PCR followed by direct sequencing [46,47]. However, the *TERTp* region around the mutations is characterized by a high GC nucleotide content (over 80%), affecting the efficacy of PCR amplification [48,49]. Despite the technical constraints, because Sanger DNA sequencing is largely deployed in many laboratories, it has been used as a simple method for *TERTp* mutation analysis in many research studies [50,51,52,53,54,55]. Table 2 summarizes the methodological characteristics of PCR tests for the detection of *TERTp* mutations in CNS tumor samples. Sanger sequencing is an economically achievable, informative test. However, this assay comprises a PCR step and possesses considerable sensitivity limits: mutations cannot be detected if the mutant allelic fraction (MAF) of a tumor sample does not exceed 15–20% (this means that at least 15–20% or more of tumor cells must harbor gene mutations). 

This technical limitation is an obstacle in the analysis of gliomas with high heterogeneity, such as recurrent glioblastomas. However, in such cases, early diagnosis (and early treatment of less heterogeneous tumors) may have a positive impact on the OS. 

Despite many disadvantages, Sanger sequencing is still commonly used for mutation detection, and many investigators rely on it to develop analytical methods for *TERTp*-mut detection. Bai and colleagues developed an accurate and rapid (less than 4 h) Sanger sequencing assay for screening *TERTp* mutations based on the human glioma cell line U251 (Table 2) [51]. Next, 147 cases of gliomas were analyzed on formalin-fixed, paraffin-embedded tumor (FFPE) samples. Accuracy was verified by real-time PCR: *TERTp*-mut sequences were detected with an analytical sensitivity of 10% mutant allele fraction [51].

Liu and colleagues developed an efficient protocol for detecting *TERTp*-mut—PCR based on the amplification-resistant mutation system (ARMS-PCR). In ARMS, a DNA polymerase can continue elongation only when the 3′-end nucleotide of the primer matches its target sequence. The authors constructed plasmids containing *TERTp* sequences and proposed a new protocol for *TERTp*-mut identification. The analytical sensitivity of this protocol reached 1% MAF versus 20% compared with conventional Sanger sequencing, which was confirmed on 124 human glioma samples [54]. In the work of Masui and colleagues, 41 human glioma tissue and 4 neoplastic adult brain tissues were analyzed by Sanger sequencing. All samples were subjected to histological and molecular genetic diagnosis, and mutations in the C228 and C250 of *TERTp* were examined in each case. As a result, mutations in the *TERTp* were detected in 21 of 41 tumor samples (51.2%) [55]. Diplas and colleagues presented an allele-specific quantitative PCR assay “GliomaDx” for detecting *TERTp* and *IDH* mutations in FFPE samples with low DNA content [56]. This method includes a multiplex preamplification step up to the allele-specific PCR stage. The analytical sensitivity of the method was 0.1% of the mutant allele fraction. Based on the sensitivity of the performed analysis, the developers put forward that GliomaDx is more than 200 times more sensitive than Sanger sequencing and can be performed within one hour.

Miki and colleagues used Sanger sequencing, pyrosequencing, and digital PCR (dPCR) to analyze *TERTp* status in 15 frozen tissue samples from primary and recurrent glioblastoma *IDH*-wt [65]. The authors showed that *TERTp* status correlated in primary and recurrent glioblastomas, but this consistency was only detected using dPCR, which was performed as a reference method of analysis. Sanger sequencing data showed a contradictory result in 7 of 15 cases. A possible reason why *TERTp*-mut was not detected by Sanger sequencing or pyrosequencing is the low percentage of tumor cells in the sample examined. The samples described above contained a large number of neoplastic cells because all recurrent tumors were subjected to intensive chemoradiotherapy after primary surgery, which may have caused massive gliosis and necrosis and masked the genetic changes in the tumor cells. Even in freshly frozen glioblastoma samples, assays of this type struggle to detect mutations [65]. These results illustrate the lack of sensitivity and quantification of Sanger sequencing and the important predominance of wild-type over mutant cells, highlighting the importance of choosing appropriate approaches capable of detecting low-level mutations. Sanger sequencing, RT-PCR, methods have also been used in the works of other authors [58,59,60,61,63,64]. The allele-specific quantitative PCR assay “GliomaDx” described in the research of Diplas and colleagues [56] showed a Limit of Detection (LOD) of 0.1% for C228T and C250T, when comparing it with the Sanger sequencing, in turn, has a detection limit of 10–20% [40,51].

Table 3 summarizes methods for detecting *TERTp* mutations, excluding PCR and Sanger sequencing. Pesenti and colleagues propose a mass spectrometric test (MassARRAY) that can identify the 1/19q co-deletion and simultaneously characterize hotspot mutations in *IDH1/2* and *TERTp* in tumor DNA. MassARRAY technology uses a laser desorption ionization time-of-flight (MALDI-TOF) mass spectrometry platform to perform multiplex genotyping with high accuracy [66].

A limitation of mass spectrometry tests is that they require an appropriate control sample to properly quantify the evaluated samples. The authors propose to use a patient’s blood sample as suitable tissue, which can be easily obtained at the time of surgery and stored prior to analysis. The cost of the assay is EUR 80 per sample [66].

#### 2.5.2. Droplet Digital PCR (ddPCR)

This method involves performing PCR in a large number of single droplets with a volume of one picoliter and measuring the fluorescence in each single droplet after PCR. ddPCR has greater specificity and sensitivity in detecting mutant sequences present in small amounts of DNA [77].

Adachi and colleagues evaluated the mutational status of *TERTp* with ddPCR using FFPE in gliomas and compared the results with Sanger sequencing [48]. As previously mentioned, Sanger DNA sequencing is a method with an acceptable level of sensitivity and accuracy in determining the presence of a mutation only if the mutant allelic fraction exceeds 10–15% of the total DNA pool. Adachi and colleagues repeated Sanger sequencing up to three times consecutively if the result of the first sequencing was different from that obtained by ddPCR analysis. If fresh-frozen tissue was used, the results of repeated Sanger sequencing were consistent with those obtained by ddPCR analysis. However, when FFPE samples were used, triplicate sequencing was often required to obtain reproducible results. For example, in the FFPE analysis of a “patient 7” sample with *IDH*-wt glioblastoma, ddPCR showed the presence of C250T mutant alleles at 37.8%. When sequencing the same sample using the Sanger method in the first and second PCR, the authors were unable to read the nucleotide peaks due to background noise, and satisfactory results were obtained only in the third sequencing [48]. In another study, many FFPE samples could not be sequenced clearly unless nested PCR was performed and the PCR products were carefully purified at each step. Therefore, in the process of repeating PCR, authors were concerned that if there was a small difference in the amplification efficiency of the wild-type allele and the mutant allele, the peak height of the mutant allele would change relative to the sequence peak of the wild-type allele, and the mutation rate percentage in the tumor sample would be overestimated or underestimated [40]. From this point of view, ddPCR, which treats each digital PCR product separately in one droplet, is more reliable in quantification. In the case of “patient 9”, the DNA isolated from FFPE samples was very short and could not be amplified by sequencing polymerases. At the same time, ddPCR analysis detected *TERTp*-mutant allele fractions with an accuracy of 1.0% *TERTp*-mutant DNA at a small initial amount of 1 ng of tumor DNA matrix, indicating high accuracy and sensitivity of this method. It took approximately 2.5 h to analyze nine glioma DNA samples. The cost of ddPCR analysis was higher than Sanger sequencing per sample. However, the cost of both methods became almost the same if 20 or more samples were analyzed simultaneously [48]. Ge and colleagues proposed a highly sensitive method for detecting *IDH1* and *TERTp* mutations based on ddPCR called “*IDH1-TERT*-mutation ddPCR” (“IT-ddPCR”) [68]. The authors determined the mutational status of *IDH1* and *TERTp* in 80 patients with gliomas using Sanger sequencing, ARMS, and IT-ddPCR in parallel. IT-ddPCR showed higher sensitivity compared to the other methods: the detection limit was 0.1% mutant DNA. Notably, there was a glioblastoma sample in which *TERTp*-mut was not detected by Sanger sequencing or ARMS, but *TERTp*-mut C250T was detected by IT-ddPCR. Thus, ddPCR shows itself to be the best method for detecting and quantifying mutant alleles and wild-type alleles in a short time with very high sensitivity, especially when compared to the traditional Sanger DNA sequencing method. 

#### 2.5.3. Next-Generation Sequencing (NGS) 

NGS covers millions of fragments in parallel, which is why it is often referred to as massive parallel sequencing. NGS analysis can be designed for different scales: whole-genome, whole-exome, or targeted sequencing. This is possible because of a properly selected and well-matched panel of genes that will selectively provide information for specific medical screening tasks. Due to the nature of the NGS method, the detection of mutations in the promoter region of the TERT gene always proceeds simultaneously with the detection of genome changes associated with other genes [69,70,71,72,73,77,78,79]. Whole-genome or -exome sequencing is an expensive, time-consuming, and technically difficult method to perform on small brain biopsy samples, as it requires a significant amount of DNA. To solve such problems, it is sufficient to create a highly specialized panel of genes that are widespread in a particular type of tumor disease, such as CNS tumors, etc. It is also possible to assemble panels for multiple marker genes for other cancers and, in general, a panel for extensive anticancer diagnostics [69,70,71,76,77,79].

At the same time, targeted NGS panels for sequencing selected genes or genetic regions can be applied to detect genetic alterations, including point mutations, insertions and deletions, copy number changes, and gene fusions, in accordance with the WHO 2021 CNS tumor classification criteria [6,7]. A distinguishing benefit of NGS technology is the ability to collect a large and diverse amount of valuable scientific or medical information on multiple genes simultaneously. The main disadvantage of NGS panels is that these methods are very expensive to use in small laboratories and with no constant flow of scientific or medical samples for mass analysis. In these cases, other approaches are more appropriate. On the other hand, this method can find wide application when used in the leading scientific and medical centers with specialization for the treatment of diseases of a certain profile, where there is a need to perform routine analysis of a larger number of samples [72].

In general, the NGS assay for routine CNS tumor screening should accomplish several specific tasks. (1) The assay should include a broad set of genes and mutation hot spots found in CNS tumors that can be tested for single-nucleotide variations, insertions, and deletions, as well as more complex genetic changes; (2) analysis must be reliable in the samples used for tumor DNA isolation, typically FFPE samples are used; (3) analysis must require a small amount of DNA for successful analysis of small stereotactic brain biopsies since more extensive tumor resection is not possible in many patients [70]. Nikiforova and colleagues developed the NGS method (“GlioSeq NGS”) to detect different types of genetic changes characteristic of CNS tumors, which can be used for small brain biopsies in adults and children in a single workflow. The sensitivity of the assay was 3–5% mutant allele fractions for single-nucleotide variants (SNVs) and 1–5% for gene fusions. Changes in *IDH1*, *TP53*, *TERTp*, *ATRX*, cyclin-dependent kinase inhibitor 2A (CDKN2A) gene, and *PTEN* were most frequently detected in high malignancy gliomas, as well as the B-Raf murine sarcoma viral oncogene homolog (BRAF) in low malignancy gliomas and histone family 3A (*H3F3A*) mutations in pediatric gliomas [70].

Higa and colleagues created an NGS panel of 48 genes for molecular diagnostic analysis of gliomas [69]. These 48 genes include genes previously proposed as diagnostically significant molecular markers of CNS tumors, namely *IDH1/2*, *ATRX*, capicua transcriptional repressor (CIC) gene, *TERTp*, *BRAF,* and 1p/19q co-deletion. The final NGS panel consisted of 1954 primer pairs covering (99.95%) coding sequences of 48 genes. The presented NGS panel combined molecular barcode technology (alternatively, a unique molecular identifier) into a single gene-specific primer-based target enrichment process with a clear distinction between false positives and true positives, resulting in both greater sensitivity and greater variant detection accuracy. Molecular barcoding technology aims to reduce the impact of enrichment and sequencing artifacts and can improve mutation detection accuracy. Sahm and colleagues developed an enrichment-based/hybrid capture NGS panel including all coding and selected intron and promoter regions of 130 genes periodically altered in CNS tumors [71]. The analysis takes five days from DNA extraction from the FFPE sample to the neuropathology report. This technique was applied to 79 control samples with known molecular alterations and to 71 samples to detect potential alterations. The concordance of NGS with results established by single biomarker methods was 98.0%, with discrepancies in one case where *TERTp*-mut was not called by NGS, and three *ATRX* mutations were not detected by Sanger sequencing. Importantly, in samples with low tumor cell counts, the NGS method was able to identify mutant alleles that traditional methods such as Sanger sequencing could not detect. Zacher and colleagues used their developed NGS panel of 20 genes exploiting 660 primer pairs for molecular diagnosis of 111 diffuse gliomas (58 fresh-frozen samples and 80 FFPE samples). Of previously known genetic changes (38 SNVs and 22 CNVs) in this tumor series, NGS analysis identified 60 of 60, corresponding to a sensitivity of 100%. The authors write that the cost of this NGS panel is in the range of cost required for individual molecular analysis of 2–3 genetic markers [77].

Since 2013, NGS analysis includes targeted genes mutated in diffuse gliomas: *ATRX*, *CIC*, *EGFR*, far upstream element-binding protein 1 (FUBP1) gene, notch receptor 1 (NOTCH1) gene, and *PTEN*; *H3F3A*, *IDH1/2*, phosphatidylinositol-4, 5-Bisphosphate 3-kinase catalytic subunit alpha (PIK3CA) gene and *BRAF*, amplifications in *EGFR* or mouse double minute 2 proto-oncogene (MDM2) and chromosome 1p, 7, 10 and 19q copy number changes are part of routine diagnosis in the Brain Tumor Center at the Cancer Institute, Rotterdam, Netherlands [72]. NGS analysis was used for all histologically diagnosed grade II and III gliomas, oligodendrogliomas, and glioblastomas in patients under 55 years of age and all difficult cases. Over four years, 433 specimens were analyzed, of which 176 cases were diagnosed with grade II/III gliomas (40.6%), and 201 cases were diagnosed with glioblastomas (46.4%) based on histologic analysis. In 123 of 433 cases (28.4%), molecular characterization led to a change in diagnosis. In 22 of the 433 patients, histology did not provide a definitive answer, but NGS analysis led to a diagnosis in 17 of these 22 cases. Moreover, in 8 of 22 cases, the pathologist did not find any evidence of a tumor, whereas NGS analysis showed the presence of glioblastoma in 4 patients, oligodendroglioma in 1 patient, and *BRAF*-mutated tumor in 1 patient. The NGS test managed to be performed on a very limited amount of tissue: the minimum requirement was 1 ng. DNA from approximately 150 cells consisted of at least 30% neoplastic cells, regardless of the method by which the tissue was obtained. However, in 19 of the 433 cases, no mutations or copy number changes were detected by NGS analysis, and in 15 of these 19 cases, a histopathological diagnosis was made. Most of them were rare brain tumors without a characteristic molecular profile that could be used to distinguish them [72].

Routine use of panel NGS to simultaneously assess multiple relevant markers is a reliable and effective method to identify novel genetic changes and classify brain tumors better than simply relying on histological examination of the tissues [69,70,71,72,76,77,78,79]. This provides clinicians with valuable information on prognosis and additional potential therapeutic options, such as target-oriented therapies, for patients with gliomas according to different racial or ethnic groups [76]. The results suggest that NGS performed on panels of genes is a promising diagnostic method that can facilitate comprehensive histologic and molecular classification of gliomas. It is expected that NGS-based molecular analysis may play an increasingly important role in formal cancer classification and treatment of brain tumors in the future [70].

#### 2.5.4. Nanopore Sequencing

Euskirchen and colleagues described a portable sequencing device that provides the same-day determination of chromosome copy number, point mutations, and methylation profiling [74]. Nanopore sequencing interprets the changes in ionic currents observed as individual DNA molecules pass through nanometer-sized protein pores. In doing so, the nanopores are able to distinguish not only nucleotides of the DNA but also modifications of individual bases, such as cytosine 5-methylation. Less tumor DNA is required to detect mutations compared to NGS or ddPCR assays. 

For convenience, we compare the main characteristics of *TERTp* mutation detection methods in Table 4.

Low-frequency whole-genome sequencing was used to simultaneously determine the chromosome copy number and methylation profiles of the original tumor DNA in the same sequencing cycle. Point mutation variants in *IDH1/2*, *TP53*, *H3F3A*, and *TERTp* were identified using deep amplicon sequencing [74].

Although recent improvements in the method have reduced read errors and the method generally has several significant advantages over NGS, there are insufficient data on the estimation of the percentage of mutant alleles using nanopore sequencing to recommend this method for use in routine clinical practice. More investigations are needed to develop a reliable analytical technique for working with targeted *TERTp* mutations.

#### 2.5.5. Magnetic Resonance Imaging (MRI)

Radiogenomics examines the relationship between radiological features and gene phenotypes. In recent years, radiomics has received much attention due to the presentation of medical images containing information on the pathophysiology and prognosis of diseases. Magnetic resonance spectroscopy (MRS) is a noninvasive magnetic resonance imaging technique that provides detailed data on cellular metabolism. The use of magnetic resonance imaging to characterize *TERTp* status is a noninvasive and efficient technique. MRI can improve the accuracy of glioma diagnosis by studying predictors of *TERT* mutations with radiomicroanalysis. Authors found that the radiomics method can well predict *TERTp* mutations and can reveal radioactive necrosis and predict recurrence and OS [80]. Ozturk-Isik and colleagues showed that proton MR spectroscopy with short echo time (1H-MRS) in 3T could be used for the noninvasive detection of *IDH* and *TERTp* mutation status. Using this method, the authors analyzed 112 diffuse gliomas. Short 1H-MRS identified the presence of an *IDH* mutation with 88.39% accuracy, 76.92% sensitivity, and 94.52% specificity, and the presence of a *TERTp* mutation in primary *IDH*-wt gliomas with 92.59% accuracy, 83.33% sensitivity, and 95.24% specificity [75].

Jiang showed that MRI-based radiomicroscopic signature is reliable for the noninvasive assessment of *TERTp* mutations in gliomas regardless of *IDH* status. Eighty-three local patients with confirmed pathology were retrospectively included as a training cohort, and thirty-three patients from the Cancer Imaging Archive (TCIA) were used for independent verification. Three types of regions of interest (ROIs), which encompassed tumor, peritumoral areas, and tumor plus peritumoral areas, were delineated on 3D contrast-enhanced T1-weighted images and T2-weighted images. Inclusion of the peritumoral area did not significantly improve the findings. The three different radiological signatures showed good accuracy and balanced sensitivity and specificity [81].

In the work of Li and colleagues, radiomics based on time-to-peak (TTP) images extracted from dynamic *O*-(2-[18F]-fluoroethyl)-l-tyrosine ([18F]-FET) positron emission tomography (PET) can predict *TERTp* mutation status in diffuse high malignancy *IDH* wild-type astrocytic gliomas with high accuracy before surgery. The TTP model included nine selected features and achieved the highest predictive ability of the *TERTp* mutation with the area under the curve (AUC) of 0.82 (95% confidence interval (CI) 0.71–0.92) and a sensitivity of 92.1% in an independent testing cohort. On the other hand, weak predictive ability was obtained in the tumor-to-background ratio (TBR) based on 5–15 min summation images model with an AUC of 0.61 (95% CI 0.42–0.80) in the testing cohort, whereas no predictive ability was observed in the TBR 20–40 model [82].

Yan and colleagues applied MRI-based radiomics to predict molecular bands noninvasively and assess their prognostic value. They retrospectively identified 357 patients with gliomas and extracted radiological features from their preoperative MRI images. Image fusion models were constructed by combining significant radiomicroscopic signatures. By separately predicting molecular markers, predicted molecular groups were obtained. Prognostic nomograms were developed based on predictive molecular groups and clinicopathological data to predict progression-free survival (PFS) and OS. Results showed that an image fusion model including radiological signatures from T1-weighted contrast-enhanced imaging (cT1WI) and apparent diffusion coefficient (ADC) achieved AUCs of 0.884 and 0.669 to predict *IDH* and *TERT* status, respectively [83].

Currently, the perfusion method plays an essential role in estimating the degree of glioma malignancy. This approach allows the evaluation of the effectiveness of histogram analysis of MRI with dynamic susceptibility contrast (DSC) and dynamic contrast enhancement (DCE) in distinguishing the states of molecular biomarkers and survival in patients with glioma. Zhang and colleagues included 43 patients with glioma for whom MRI with DCE and DSC was performed. Relevant molecular test results were collected, including *IDH*, *MGMT*, and *TERT*. Differences in each parameter between gliomas with different expression states (*IDH, MGMT*, and *TERT*) were evaluated. In addition, the diagnostic efficacy of each parameter was analyzed. OS of all patients was assessed. The DCE-MRI histogram demonstrates high diagnostic efficiency in identifying different molecular types and prognostic gliomas evaluation [84].

Fukuma developed a method to predict tumor genotypes using a pre-trained convolutional neural network (CNN) from magnetic resonance (MR) images and compared the accuracy with that of a diagnosis based on conventional radiological characteristics and patient age. Multisite preoperative MR images of 164 patients with grade II/III glioma were grouped by *IDH* and *TERTp* mutations as follows: (1) wild-type *IDH*, (2) *IDH* and *TERTp* co-mutations, (3) mutant *IDH* and wild-type *TERTp*. A CNN (AlexNet) was applied to the four types of MR sequences, and CNN texture characteristics were obtained for group classification using a linear reference vector machine. The classification was also performed using common radiological features and/or patient age. Using all features, they were able to classify patients with an accuracy of 63.1%, which is considerably higher than the accuracy obtained using only radiological features or patient age [85].

In retrospective research, Ivanidze and colleagues included 29 patients with *IDH1/2* wild-type glioblastoma (13 *TERT*-wt, 16 *TERT*-mut) who underwent preoperative imaging MRI. Qualitative image phenotypes were assessed using the Visually Accessible Rembrandt Images (VASARI) feature set. In addition, histograms of ADC and DCE MR perfusion values were analyzed with increasing tumor volumes of interest, and differences between wild-type *TERT* tumors and tumors with a *TERT* mutation were assessed. The research demonstrates altered permeability rates associated with the *TERT* mutation in glioblastoma, laying the foundation for future studies evaluating implications for therapeutic treatment and clinical outcomes [86].

This direction is attractive because advanced MRI techniques are playing an increasingly important role in the clinical treatment of glioma by new MRI options to maximize resection safety, minimize surgical risk, individualize the treatment plan, and prolong patients’ lives. In addition, the introduction of MRI with advanced morphological, functional, and metabolic imaging capabilities increasingly makes comprehensive glioma diagnosis possible using machine learning and neural network techniques [87].

### 2.6. Specifics of Mutation Detection in the TERTp According to the DNA Source

Currently, molecular genetic testing of tumor tissues obtained by biopsy before or during therapy when tumor resistance may develop is the gold standard in clinical and research practice. Usually, DNA is taken from frozen tumor material or FFPE for performing genetic tests. FFPE samples are widely used for molecular detection because they are suitable for long-term storage. However, FFPE samples can lead to deamination of cytosine residues, DNA degradation, and single-nucleotide substitutions [36,88]. Bai and colleagues evaluated FFPE and frozen tissue samples for the presence of *TERTp* mutations [51]. The detection rate of *TERTp* mutations in FFPE samples was slightly lower compared to frozen tissues (37.0% vs. 42.0%, respectively), possibly because DNA damage and base changes occur more frequently in FFPE samples than in frozen samples.

A tissue biopsy can poorly represent tumor heterogeneity, and most tumors, especially in later stages, are characterized by multiple heterogeneity. In addition to intratumor heterogeneity, there is intermetastatic heterogeneity, that is, even metastases from the same patient have a different molecular genetic profile. Analysis of biopsy or operative material obtained from one part of a tumor cannot provide a complete understanding of the tumor’s genetic profile and its metastases. Furthermore, biopsies of different parts of a tumor in stages are quite time-consuming and it is expensive to analyze each tumor section taken. In addition, biopsies are invasive, and the patient’s condition or tumor location may be dangerous and/or unacceptable, especially for elderly patients and/or patients with poor general health. In this context, the concept of liquid biopsy has emerged in neuro-oncology, based on the molecular characterization of freely circulating tumor DNA found in easily accessible fluids such as plasma or cerebrospinal fluid [4,44].

Liquid biopsy is an alternative way to investigate the molecular profiles of tumors, in which blood, cerebrospinal fluid, urine, and other body fluids are used to detect and isolate circulating tumor cells and circulating tumor DNA (ctDNA) in cancer patients. The biological basis for this approach is that dying cells (necrosis, apoptosis) release some of their DNA into the systemic bloodstream (or other body fluids), so tumor-specific mutated DNA can be detected by highly sensitive methods such as ddPCR and NGS, provided that tumor DNA fragments are in sufficient quantity for analytical detection in the bloodstream (or other body fluids) of cancer patients. Unfortunately, given the relatively low content of ctDNA compared to non-tumor cell-free (cfDNA) in the bloodstream, standard approaches such as Sanger sequencing or pyrosequencing are only able to detect mutant tumor DNA fragments in patients with significant tumor burden and high levels of ctDNA in biological fluids. The release of ctDNA into blood and body fluids depends on the location and morphology of the tumor, the size of the malignancy, and the degree of vascular infiltration of the tumor.

The ctDNA fragments contain genetic defects identical to those present directly in the tumor. Therefore, the ctDNA analysis can provide the same genetic information as the biopsy samples, despite the specific localization of the CNS tumor [73]. The possibility to analyze blood has distinct advantages. Blood sampling is a minimally invasive procedure and avoids complications arising from the biopsy. In addition, blood can be drawn from the patient at any time of medical observation throughout the course of the prescribed therapy. It is also possible to evaluate the efficacy of a drug or other therapy as the amount of ctDNA released increases at different stages of the therapy.

ctDNA, which contains somatic changes from the tumor, has proven to be a marker for disease monitoring. In research conducted by Fontanilles and colleagues on 52 patients with glioblastoma who received temozolomide and radiotherapy simultaneously, followed by a phase of maintenance radiotherapy, an increase in the concentration of ctDNA (determined by the presence of mutations in the *TERTp*) was observed over time from the start of treatment (taking temozolomide and radiotherapy) to the beginning of disease progression, from 9.7 to 13.1 ng/mL, respectively, while no difference was observed for non-progressive patients [89]. The frequency of the detection of cfDNA in the plasma of the patients with gliomas varies in different research from 7.9 to 55%. *TERTp* mutations in cfDNA were detected by ddPCR in two patients (3.8%) and only in gliosarcoma subtypes, whereas 85% of tumor tissue samples were *TERTp* mutated [89]. The failure to detect *TERTp-mut* in plasma may be due to the small size of the ctDNA fragments (<70 bp), which modern sequencing methods cannot detect. The blood–brain barrier between the circulatory and central nervous systems physically limits the crossing of large DNA fragments. According to Muralidharan and colleagues, the overall sensitivity and specificity for the detection of *TERTp-mut* in cfDNA from plasma of patients with glioma by ddPCR was 62.5% and 90%, respectively, compared with DNA from FFPE samples in matched tumor tissues [67].

Cerebrospinal fluid as a source of ctDNA for glioma analysis proved to be better than plasma [90]. Juratli and colleagues evaluated the ability to detect *TERTp-mut* in cerebrospinal fluid cfDNA and plasma ctDNA of patients with glioblastoma. Thirty-eight patients had *TERTp*-mutant/*IDH* wild-type glioblastomas. The matched *TERTp-mut* in cerebrospinal fluid ctDNA was successfully detected with 100% specificity and 92.1% sensitivity. On the other hand, the sensitivity of *TERTp-mut* detection in plasma ctDNA was significantly lower—7.9% [90]. This fact may be explained by the existence of the blood–brain barrier, which limits the release of CNS tumor ctDNA into the bloodstream. In the work of Miller and colleagues, cerebrospinal fluid samples collected from 85 patients who developed new neurological symptoms after treatment of diffuse glioma were examined by NGS for the presence of mutations, copy number changes, and chromosomal aberrations. Tumor DNA was detected in the cerebrospinal fluid of 42 patients (49.4%) by at least one tumour-derived genetic alteration and contained WHO 2016 recommended glioma markers, including 1p/19q co-deletions, *IDH1* and sub-type glioma markers *TP53*, *ATRX* and *TERTp* mutations, as well as *CDKN2A* and/or *CDKN2B* deletions, and *EGFR* aberrations and others with low frequency [91]. This research shows that tumor DNA from the cerebrospinal fluid of patients with glioma provides a complete and genetically reliable representation of the tumor genome.

Summarizing the recent research, we would like to emphasize that molecular genetic analysis of plasma or cerebrospinal fluid ctDNA for the detection of malignant tumor-induced mutations is a promising approach for managing patients with CNS tumors regarding grading and treatment prognosis. However, the development of new and highly sensitive methods for the detection of ctDNA is necessary. Moreover, techniques that use ctDNA as a sample for CNS tumor lesions are critically limited (Table 3). At the same time, methods for the noninvasive analysis of *TERTp* mutation status have been developed and successfully applied for other types of malignancies, for example, in urine DNA samples to detect urothelial tumors [92]. In this research, the authors developed the technique NGS UroMuTERT, which costs EUR 15 per assay (one single amplicon that also captures C228A and CC242-243TT mutations in addition to C228T and C250T). In another research study dedicated to the development of sensitive ddPCR assays for detecting urinary *TERTp* mutations as noninvasive biomarkers of urothelial cancer, Sanger sequencing costs EUR 8 per strand, so forward and reverse strand costs EUR 16, representing two *TERTp* mutations (C228T, C250T) in the amplicon; ddPCR costs EUR 12 per assay (EUR 6 per mutation in *TERTp*, and if two more rare mutations C228A and CC242-243TT are included in the assay, it would cost EUR 24) [93]. 

The prices in the article are approximate and were calculated on the experimental work of the authors at the time of publication. 

## 3. Conclusions

Currently, molecular genetic classification of cancer diseases increasingly enters the daily clinical practice for the treatment of patients owing to the possibility of obtaining information for timely diagnosis, prognosis information during the course of the disease, and information for personalized treatment strategies. These additional molecular tests to histology are needed to make an accurate diagnosis. However, the inclusion of molecular markers in routine diagnostics requires their rapid and reliable assessment. The lack of this assessment complicates the robust validation of the results obtained and prevents timely decisions regarding the clinical management of patients with CNS tumors with the selection of the most appropriate treatment protocols. Unfortunately, a potential biomarker that could be detected by high-throughput, cost-effective and easy to implement technologies for the rapid, reliable, and inexpensive diagnosis of CNS has not been found yet. Nevertheless, some solutions may be appropriate and reasonable at the expense of more patients.

The review discusses the possibility of focusing on testing changes in the mutational status of the TERT gene promoter using various existing techniques. A comparison of the analytical sensitivity of the different methods is complicated by the various methods of DNA isolation, as well as the heterogeneity of tumor samples and the often limited and unrepresentative selection of patients. However, some conclusions can be drawn.

The main advantage of the NGS-based approaches is the simultaneous detection of other markers relevant for glioma diagnosis in addition to mutations in the TERT gene promoter, but their high cost, complicated workflows, and need for highly skilled personnel in both generating and analyzing data limit their widespread use in the clinic. 

It should also be noted that significant progress in the noninvasive diagnosis of *TERT* promoter mutational status is possible with the use of new MRI approaches for the analysis of medical image data containing information on pathophysiology and disease prognosis. This method can provide additional support to the methods of molecular genetic analysis of glioma biomarkers for more accurate diagnosis. However, this method requires further development and integration with other methods to work effectively together.

Sanger sequencing is an economically feasible test, but mutations cannot be detected if less than 15–20% of sample cells carry somatic changes in the target gene. Quantitative PCR techniques can overcome this limitation, but standardization of the methods depending on the DNA source is necessary. However, digital PCR methods combine the advantages of high sensitivity and accuracy with low starting amount of materials and have proved to reliably detect mutations in circulating fluids, highlighting their promising future for implementation in routine clinical practice.

## Figures and Tables

**Figure 1 biomedicines-10-00728-f001:**
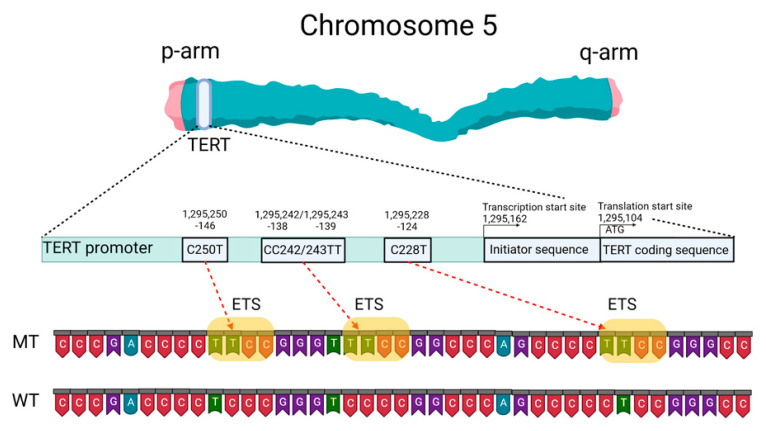
Schematic presentation of TERT gene at chromosome 5p, its promoter structure and two canonical mutations causing gliomagenesis. C > T mutation occurs at one of both positions of the *TERTp* (−124 and −146 to ATG for C228T and C250T, respectively) in gliomas, which create de novo ETS binding motifs. CC242/243TT is a rare mutation and has not previously been seen in gliomas; it has been observed in other types of cancer. The figure was created with BioRender.com (19 March 2021).

**Table 1 biomedicines-10-00728-t001:** Frequency of *TERT* mutations in different types of gliomas (type of mutation *TERTp*: C228T and C250T, respectively).

	Authors:	Arita et al. [19]	Heidenreich et al. [22]	Yuan et al. [32]	Arita et al. [33]	Pekmezci et al. [34]	Yang et al. [35]	Kim et al. [36]	You et al. [37]	Huang et al. [38]
Diagnosis:										
Diffuse astrocytoma		19%			29%			33%		20%
Anaplastic astrocytoma		25%			33%		33%	30%	32%	33%
Astrocytoma			39%	24%		22%	7%		11%	
Glioblastoma		70%	80%	69%	58%	66%		64%	42%	84%
Oligoastrocytoma		36%		38%	49%		54%		54%	
Anaplastic oligoastrocytoma		40%			44%		42%		41%	
Oligodendroglioma		74%	70%	72%	83%	96%	74%		76%	70%
Anaplastic oligodendroglioma		74%			74%		67%	100%	53%	
Number of patients in the research		546	303	3477	758	1208	377	67	684	204

**Table 2 biomedicines-10-00728-t002:** Accuracy and methodological characteristics of tests to detect *TERTp*-mut in CNS tumor samples.

Method, Patient Group	Type of Tumor, Number of Patients	Material of Tumor DNA	MAF, %	Primers on *TERTp* Sequence (forward and Reverse)	PCR Product Length, n.b. (If Specified)	Reference
Group 1: Pyrosequencing group 2: PCR and Sanger sequencing	Glioma (1) 242/304 (2) 127	Frozen tissue samples	dnp *	Primers to amplify the region including both sites: 5′-TCCCTCGGGTTACCCCACAG-3′ and 5′-AAAGGAAGGGGAGGGGCTG-3′ (biotinylated at the 5′-end) The primer for pyrosequencing (5′-ACCCCGCCCCGTCCCGACCCC-3′) was constructed above C250T, so the two hotspots are analyzed in the same assay with the dispensation order 5′-GTCGTCCGCATGCCTC-3′ to pyrosequence 5′-TT/CCCGGGTCCCCGGCCCAGCCCCT/CTCCG-3′ (AQ PyroMark Q96 (version 2.5.7) analysis, applied to underlined positions) 5’-TCCCTCGGGTTACCCCACAG-3′ and 5′-AAAGGAAGGGGAGGGGCTG-3′	356 **	[19]
PCR and Sanger sequencing	Gliomas, 325	Frozen tissue	dnp	5′-CCCACGTGCGCAGCAGGAC-3′ and 5′-CTCCCAGTGGATTCGCGGGC-3′	260	[22]
Nested PCR and Sanger sequencing	Glioblastoma, 358/32	FFPE samples	20%	1st PCR: 5′-GTCCTGCCCCTTCACCTT-3′ and 5′-GCACCTCGCGGTAGTGG-3′ Nested PCR: 5′-CCGTCCTGCCCCTTCACC-3′ and 5′-GGGCCGCGGAAAGGAAG-3′ For samples which were not amplified, used set 3: 5′-TTCCAGCTCCGCCTCCT-3′ and 5′-GCGCTGCCTGAAACTCG-3′	273 128 145	[40]
Reverse transcription PCR (“RT-PCR”) and Sanger sequencing	Gliomas class II, III and IV (1) group 235 (2) group 897 Total control group 1090	(1) Blood (2) Frozen tissue and FFPE samples	dnp	*TERTp* was amplified using 5′-GGCCGATTCGACCTCTCT-3′ (5′-GTCCTGCCCCTTCACCTT-3′ for FFPE samples) and 5′-AGCACCTCGCGGTAGTGG-3′	489 **	[52]
PCR and Sanger sequencing	Gliomas of low malignancy, 237	FFPE samples	dnp	5′-GTCCTGCCCCTTCACCTT-3′ 5′-CAGCGCTGCCTGAAACTC-3′	163	[53]
Nested PCR and Sanger sequencing	Grade III gliomas, 377	FFPE samples	dnp	1st PCR: 5′-GTCCTGCCCCTTCACCTT-3′ and 5′-GCACCTCGCGGTAGTGG-3′ Nested PCR: 5′-CCGTCCTGCCCCTTCACC-3′ 5′-GGGCCGCGGAAAGGAAG-3′	273 128	[35]
Nested PCR and Sanger sequencing	Glioma, 887	Tissue frozen in liquid nitrogen (80% of tumor cells)	dnp	1st PCR: 5′-GTCCTGCCCCTTCACCTT-3′ 5′-GCACCTCGCGGTAGTGG-3′ Nested PCR: 5′-CCGTCCTGCCCCTTCACC-3′ 5′-GGGCCGCGGAAAGGAAG-3′	273 128	[37]
Sanger sequencing ddPCR	Glioma, 9	Frozen tissue and FFPE samples	C228T C250T 1.0%	5′–TCCCTCGGGTTACCCCACAG–3′ and 5′–AAAGGAAGGGGAGGGGCTG–3′	356 **	[48]
PCR and Sanger sequencing	Glioma, 147	Frozen tissue and FFPE	C228T 10%	M13F: 5′-AGTGGATTCGCGGGCACAGA-3′ and M13R: 5′-CAGCGCTGCCTGAAACTC-3′ Primer M13: 5′-TGTAAAACGACGGCCAGT-3′ and 5′-CAGGAAACAGCTATGACC-3′	235	[51]
PCR and Sanger sequencing	Glioma, 41/4	FFPE samples	dnp	5′-TCCCTCGGGTTACCCCACAG-3′ and 5′-AAAGGAAGGGGAGGGGCTG-3′	356 **	[55]
Allele-specific quantitative PCR assay “GliomaDx”	39 diffuse glioma	Frozen tissue	0,1% MAF, or 0.2% tumor cells	5′-CAGCGCTGCCTGAAACTC3′ and 5′-GTCCTGCCCCTTCACCTTC-3′	163 **	[56]
Pyrosequencing	Glioma, 179	Tumor tissue (>80% tumor cells)	dnp	5′-GTCCTGCCCCTTCACCTT-3′ and 5′-GCACCTCGCGGTAGTGG-3′ (both biotinylated at the 5′ end) Primer for pyrosequencing: 5′-TGTAAAACGACGGCCAGT-3′ 5′-CAGGAAACAGCTATGACC-3′ (both biotinylated at the 5′ end)	273 **	[57]
PCR and Sanger sequencing	Glioma, 124	FFPE samples (>50% of tumor cells)	dnp	5′-AGCACCTCGCGGTAGTGG-3′	dnp	[58]
PCR and Sanger sequencing	Primary CNS tumors, 301	FFPE samples and blood (ctDNA and cfDNA)	dnp	5′-GTCCTGCCCCTTCACCTTC-3′ and 5′-AGCACCTCGCGGTAGTGG-3′	274	[59]
PCR and Sanger sequencing	Primary glioblastoma, 67	FFPE samples	dnp	5′-GTCCTGCCCCTTCACCTT-3′ and 5′-CAGCGCTGCCTGAAACTC-3′	163 **	[60]
Chip-based dPCR system	Diffuse glioma, 34	Samples of cerebrospinal fluid	dnp	dnp	dnp	[61]
Sanger sequencing	Glioma, 168	FFPE samples	dnp	5′-M13-GTAAAACGACGGCCAGTCACCCGTCCTGCCCCTTCACCTT-3′ (M13: 5′-GTAAAACGACGGCCAGT-3′ and 5′-GCACCTCGCGGTAGTGG-3′)	300–310	[62]
Sanger sequencing	Glioma, 200	FFPE samples	dnp	5′-CACCCGTCCTGCCCCTTCACCTT-3′ and 5′-GGCTTCCCACGTGCGCAGCAGGA-3′.	193 **	[42]
Sanger sequencing	Glioma, 444	FFPE samples	dnp	5′-GCACAGACGCCCAGGACCGCGCT-3′ and 5′-TTCCCACGTGCGCAGCAGGACGCA-3′	196	[63]
RT-PCR	Glioma, 1208	FFPE samples	dnp	5′-AGTGGATTCGCGGGCACAGA-3′ and 5′-AGCACCTCGCGGTAGTGG-3′	346	[36]
Sanger sequencing	Glioma, 15	FFPE samples	dnp	5′-CAGCGCTGCCTGAAACTC-3′ and 5′-GTCCTGCCCCTTCACCTT-3′	163 **	[64]

dnp *—data not provided; LOD: Limit of Detection; MAF: mutant allelic fraction. **—The length of the amplicon was calculated by us.

**Table 3 biomedicines-10-00728-t003:** Diagnostic characteristics of different methods for detecting *TERTp* mutations.

Method	Detectable Markers	Material of Tumor DNA	Type of Tumor, Number of Patients	Limit of Detection	Sensitivity	Reference
ddPCR	*TERTp* mutations	Fresh-frozen samples and FFPE samples	9 gliomas	1% mutant DNA	dnp	[48]
MassARRAY Mass spectrometry	1p/19q co-deletion mutations *TERTp* and *IDH*	FFPE samples (tumor cell content in all samples was at least 70%)	50 gliomas	dnp *	dnp	[66]
ddPCR	*TERTp* mutations	Plasma cfDNA	157 gliomas	0.27% (C250T) and 0.42% (C228T).	62.5%	[67]
*IDH1*-*TERT*-mutation ddPCR (IT-ddPCR)	Mutations *TERTp* and *IDH1*	FFPE samples	80 gliomas	0.1% mutant DNA	dnp	[68]
NGS analysis	Analyzes 48 genes including *TERT*	FFPE samples	106 gliomas	dnp	dnp	[69]
NGS analysis (GlioSeq)	Analyzes 68 genes including *TERT*	Frozen tissue and FFPE samples from surgically removed CNS tumors	54 tumors of CNS	3–5% mutant alleles for SNV and 1–5% for gene fusions.	100%	[70]
NGS analysis	Analyzes 130 genes including *TERT*	FFPE samples	150 CNS tumors	dnp	99.0%	[71]
NGS analysis	*ATRX, CIC, EGFR, FUBP1, NOTCH1, PTEN, H3F3A, IDH1/2, PIK3CA,* and *BRAF*, amplification of *EGFR, MDM*, chromosome copy number changes 1p, 7, 10 and 19q	FFPE samples	433 diffuse gliomas	dnp	dnp	[72]
NGS analysis Guardant360 test	54, 68, 70 and 73 genes including *TERT*	ctDNA	419 primary brain tumors	dnp	dnp	[73]
Sequencing 3rd Generation (Nanopore)	*IDH1, IDH2, TP53, H3F3A*, and the *TERT*	Fresh-frozen tumor tissue	28 CNS tumors	dnp	dnp	[74]
Magnetic resonance imaging	*TERTp* and *IDH* mutations	Not applicable	112 diffuse gliomas	dnp	83.33%	[75]
Multigene (NGS) panel	*TP53, TERT, IDH1, PTEN, ATRX, EGFR,* and others	FFPE samples	81 gliomas	dnp	dnp	[76]
Multigene (NGS) panel	*ATRX, BRAF, CDKN2A, CDKN2B, CDKN2C), CIC, EGFR, FUBP1, H3F3A, IDH1, IDH2, NF1, NF2, NRAS, PIK3CA, PIK3R1, PTEN, RB1, TERTp,* *TP53*	58 fresh-frozen samples and 80 FFPE samples	121 diffuse gliomas	dnp	100%	[77]
Multigene (NGS) panel	*TERT, IDH1, TP53, PTEN, NOTCH1, EGFR,* and others	Fresh-frozen samples	81 gliomas, 303 glioblastomas, 509 lower-grade gliomas	dnp	dnp	[78]
Multigene (NGS) panel	*ATRX, BRAF, CDKN2A, CDKN2B, CDKN2C), CIC, EGFR, FUBP1, H3F3A, IDH1, IDH2, NF1, NF2, NRAS, PIK3CA, PIK3R1, PTEN, RB1, TERT*p, *TP53*	FFPE samples	345 diffuse gliomas	dnp	dnp	[79]

dnp *—data not provided.

**Table 4 biomedicines-10-00728-t004:** Comparison of characteristics of methods for determining *TERTp* mutations.

The Method	The Amount of Material Required	Number of Mutations Determined Simultaneously	Sensitivity of the Method	Method Specificity	Cost of Analysis	Technical Complexity	Time of Analysis	Pros, Cons and Limitations
ddPCR	Low (1–5 ng of DNA)	One	1%	High	High	High	During the working day	+: Low amount of the DNA template; absolute quantification; resistance to PCR inhibitors −: High cost of assays; need for well-trained staff; higher contamination risk; limited and defined targets
Multiplexed dPCR	Low (1–5 ng of DNA)	Several	1–2%	High	Medium	Medium	During the working day	+: Low amount of the DNA template; feasibility; absolute quantification; resistance to PCR inhibitors; user-friendly system −: Considerable cost of analyses; high contamination risk; limited and defined targets
dPCR	Low (1–5 ng of DNA)	One	1%	High	Medium	Medium	During the working day	+: Low amount of the DNA template; feasibility; absolute quantification; resistance to PCR inhibitors; user-friendly system −: Considerable cost of analyses; single-target method; high contamination risk; limited and defined targets
NGS mutation panel	Moderate (5–10 ng of DNA)	A lot of	1 to 5%	High	High	Very high	In a few days	+: Satisfactory estimate of MAF; availability of diverse commercial tests; possibility of detecting large diversity of targets including unpredictable mutations and allelic forms −: High cost of assays; time-consuming method
Nanopore sequencing	Very low (1 ng of DNA)	A lot of	1%	High	High	High	Within 6 h	+: Possibility of detecting large diversity of targets including unpredictable mutations and allelic forms −: High error rates; high cost
Sanger sequencing	High (10 ng of DNA)	Several	10 to 20%	Medium	Low	Low	Within 4–6 h	+: Low cost −: High error level; high Limit of Detection; quantitative tests are problematic

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
