# Peer review of "Detection of TERT Promoter Mutations as a Prognostic Biomarker in Gliomas: Methodology, Prospects, and Advances"

_biomedicines, 2022, doi:10.3390/biomedicines10030728_

Round 1

Reviewer 1 Report

Dear authors

The authors submitted a review manuscript entitled, "Detection of TERT promoter mutations as a prognostic biomarker in gliomas: methodology, prospects, and advances." It’s an interesting issue. The TERT promoter mutations have been well-discussed and reviewed in quite a few recent articles, although its advances are still of importance. The mutational status of the telomerase reverse transcriptase (TERT) promoter has been widely investigated in various tumors, including gliomas. The authors aimed to give an integrative review with recent updates. The article is well-written and logically arranged. However, there are major concerns on this review article. 

  1. There are quite a few similar review articles recently published. For example, J Cancer Res Clin Oncol. 2021 Apr;147(4):1007-1017; Int J Mol Sci. 2021; 22(19):10373; and Brain Tumor Pathol. 2020 Apr;37(2):33-40. Some of them were listed in the references. The authors should emphasize the differences and the uniqueness of this review. New insights and new updates may be highlighted.
  2. In the session of "Analysis of methods to detect TERT promoter mutations in CNS tumors", the comparison between different methods can be the uniqueness of this review, but an integrative table or figure to show their pros and cons & limitation were not seen.
  3. Although the authors listed the molecular results of cohorts with gliomas, the diagnostic sensitivity varies in table 3. In table 3, the column of "Limit of detection (sensitivity)" showed inconsistent data. For example, "1% mutant DNA" may be the limit, but the sensitivity was not shown. And "0,27% (C250T) и 0,42% (C228T)" was a misleading typing.
  4. Gliomas are composed of tumor cells that histologically show varying degrees of atypia and aggressiveness. The molecular profiles between how and high grade gliomas or different locations may have different mutation rates. The authors may need to provide the discussion of different TERT mutation rates or patterns in different histological grading and location. 
  5. The author contribution should be explained in detail; otherwise, we cannot know why the first and corresponding authors have a significant contribution. Who did the visualization (figures, tables, etc), thorough editing, and major conceptualization should be clarified. 

Author Response

Dear reviewer, thank you very much for your valuable comments for our work!

Your comments have been worked through and incorporated by us into the corrected version of the manuscript.

Some comments required clarification from our side, and will be listed point by point according to your suggestions.

Point 1. We focus primarily on the technical side of detection methods rather than on the significance of certain biomarkers in molecular tumor genotyping and their importance in assessing disease progression status, as in these reviews. In our review, we have tried to review all existing methods for determining TERTp mutational status in gliomas that have appeared in the last 5-10 years, taking into account current advances in the development of new research techniques. We consider both traditional methods, such as Sanger sequencing, and new methods, such as digital PCR, droplet digital PCR, NGS, and nanopore sequencing. Magnetic resonance methods and other rare approaches such as mass spectrometry are also discussed. All methods are considered in terms of their application to determine the mutational status of the TERTp mutations in central nervous system tumors. We focus the review specifically on one key biomarker of TERTp and all the technical possibilities for its determination. These methods are also compared with each other. This is the main difference between this review and the ones you refer to.

Point 2. In this review, the methods are compared with each other in Table 4. We have added a comparison of the pros, cons, and limitations of the different methods additionally in Table 4 to make it easier to understand and comprehend the data presented. We think that Table 4 provides sufficient information about each method under consideration and allows for a general comparison between them. In addition, the pros and cons and limitations of the methods are discussed sequentially in the text of the review article as all methods are discussed. Please take a look at this part of the review.

Point 3. The error was corrected and the data on the sensitivity of the method was added to the table. A separation of the detection limit and sensitivity graphs was carried out. We provided the data that were presented in the articles. Most articles had no detection limits and specificity data at all, this is especially apparent for articles with the NGS method. They were rarely available in the same article. Please take a look at this part of the review.

Point 4. Before Table 1, we added in the paragraph related to the description of the percentage presence of TERTp mutations, additional data on the degree of disease according to the latest WHO classification. We have also added average percentages to characterize the relationship between presence of TERTp mutations and degree of disease for clarity. Based on the data presented above, we can conclude that the presence of TERTp mutations has no clear relationship with the degree of gradation of the tumor disease but depends more on the tumor disease directly. This is the reason why we did not combine the data in order to identify clear correlations before. Please take a look at this part of the review.

Point 5. The author contributions to the review were entered in the author contributions column. Please take a look at this part of the review.

Best regards, Tsimur Hasanau

Reviewer 2 Report

The review manuscript summarized diagnosis and prognosis value TERTp mutations in CNS diseases. From basic mechanism of pathology to detection methods, the content of the manuscript is generally comprehensive and well-prepared. I would suggest minor revisions regarding the manuscript:

  1. References are missing in the 1st paragraphs of Sec. 2.1. There is no citation in the whole paragraph.
  2. Section 2.5.3 lacks citations in general.

Author Response

Dear reviewer, thank you very much for your valuable comments for our work!

Your comments have been worked through and incorporated by us into the corrected version of the manuscript.

Some comments required clarification from our side, and will be listed point by point according to your suggestions.

Point 1. The necessary references have been added to the text. Please take a look at this part of the review.

Point 2. Table 3 summarizes all the works we found with this method as applied to the determination of TERTp mutational status in central nervous system tumors with key characteristics and references to original works. Additional references to this chapter have been added. Please take a look at this part of the review.

Best regards, Tsimur Hasanau

Round 2

Reviewer 1 Report

Dear authors

The authors submitted a revised version of a review manuscript entitled, "Detection of TERT promoter mutations as a prognostic biomarker in gliomas: methodology, prospects, and advances." The article is well-written and logically arranged. Although TERT promoter mutations have been well-discussed and reviewed in quite a few recent articles, the authors have pointed out their uniqueness point by point in their revision. After reading the revision, no obvious problem or special issue is found. Herein, no further comments on this review article are given.